# CLAP: Unsupervised 3D Representation Learning for Fusion 3D Perception via Curvature Sampling and Prototype Learning

**Runjian Chen**[1]  **Hang Zhang**[2]  **Avinash Ravichandran**[2]  **Hyoungseob Park**[3]
**Wenqi Shao**[4]  **Alex Wong**[3]*  **Ping Luo**[1,5]*

[1]The University of Hong Kong  [2]Cruise  [3]Yale University
[4]Shanghai AI Laboratory  [5]HKU Shanghai Intelligent Computing Research Center
{rjchen, pluo}@cs.hku.hk   alex.wong@yale.edu

## Abstract

Unsupervised 3D representation learning reduces the burden of labeling multimodal 3D data for fusion perception tasks. Among different pre-training paradigms, differentiable-rendering-based methods have shown most promise. However, existing works separately conduct pre-training for each modalities due to computational costs of processing large point clouds with images. As such, mutual benefit of high-level semantics (from image) and 3D structure (from point cloud) has not been exploited. To address this gap, we propose a joint unsupervised differentiable-rendering-based pre-training method for images and point clouds, termed CLAP, short for **C**urvature samp**L**ing and le**A**rnable **P**rototype. Specifically, our method overcomes the computational hurdle by Curvature Sampling to select the more informative points/pixels for pre-training. To uncover the performance benefits brought by their complementarity, we propose to use learnable prototypes to represent parts of the 3D scenes in a common feature space and an Expectation-Maximization training scheme to associate embeddings of each modality to prototypes. We further propose a swapping prediction loss that explores their interplay through prototypes along with a Gram Matrix Regularization term to maintain training stability. Experiments on NuScenes and Waymo datasets show that CLAP achieves up to $100\%$ more performance gain as compared to previous SOTA pre-training methods. Codes and models will be available here.

## 1 Introduction

3D perception facilitates spatial applications such as autonomous driving. The autonomous system, on which these applications are deployed, are typically equipped with multiple sensors including visual ones like cameras that produce RGB images, and range sensors like LiDAR (Light-Detection-And-Ranging) that generate point clouds. Fusion of these two modalities (Liu et al., 2023; Liang et al., 2022; Li et al., 2022b; Liang et al., 2019; Li et al., 2022a; Meyer et al., 2019; Chen et al., 2017) have generally improved over use of a single modality, e.g., camera (Ding et al., 2020; Chen et al., 2016; Simonelli et al., 2019; Wang et al., 2021b; Zhang et al., 2021; Ku et al., 2019) or LiDAR (Shi et al., 2019; Yan et al., 2018; Shi et al., 2021; Yin et al., 2021; Fan et al., 2021; Bai et al., 2022) separately, in 3D perception performance.

However, training multimodal 3D perception models is expensive as labeling in 3D space is notoriously time-and-energy-consuming. Unsupervised 3D representation learning, which pre-trains backbones without any label and fine-tuned the pre-trained weight for downstream performance improvement, has shown the potential to alleviate the labeling burden in 3D perception. Amongst the many unsupervised 3D pre-training methods (Liang et al., 2021; Huang et al., 2021; Chen et al., 2022; Yang et al., 2023; et al, 2023; Huang et al., 2023; Zhu et al., 2023; Yang et al., 2024), use of mask auto-encoding (reconstruction) and differntiable rendering has emerged as the most performant. However, encoding and processing high-dimensional multi-modal data (images and point

---

*Corresponding authors.

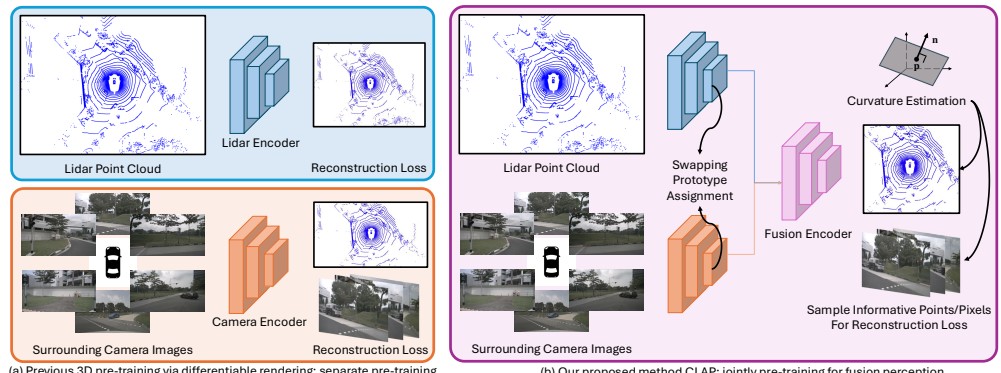

Figure 1: Unlike previous SOTA unsupervised 3D representation learning method UniPAD (Yang et al., 2024) that separately pre-train LiDAR and camera encoders with differential rendering (a), our proposed method CLAP conducts joint pre-training for fusion perception.

clouds) is computationally expensive: If one were to pre-train using all points and pixels within the point cloud and image, even the most advanced GPU to date is only able to hold a batch size of 1. Therefore, existing methods, e.g., UniPad (Yang et al., 2024), has conventionally pre-trained each modality separately. The limitation of separate pre-training is that each encoder is restricted to its own modality. Recovering 3D information from images is an ill-posed problem; point clouds provide geometry cues but lack higher-level semantics.

To address this pain point, we propose a joint unsupervised pre-training method of image and point cloud modalities based on differentiable rendering to exploit their complementarity. Our method, CLAP, short for **C**urvature samp**L**ing and le**A**rnable **P**rototype, addresses the primary computational challenge by a curvature sampling strategy, which comes from the observation that there exists redundancy in information when sampling multiple points on the same flat surface. This is enabled by estimating the curvature of each point in the 3D space by taking second order derivative of the SDF (signed distance field) function. This sampling strategy captures variations in the point cloud and in turn provide more informative points than the conventional strategy in (Yang et al., 2024).

As our curvature sampling strategy has reduced the computational burden to allow both modalities to be simultaneously processed, we propose to model the interplay between image and point cloud modalities through a set of learnable prototypes, which represents parts of the 3D scene and enables a common feature space to bridge the two modalities. These prototypes are trained via an Expectation-Maximization (EM) algorithm that maximizes similarity between embeddings for each modality and the set of prototypes. Furthermore, we propose to use the swapping prediction loss to explore the interaction between the two modalities. Last but not least, we utilize a Gram Matrix Regularization term to minimize similarity across prototypes, avoiding collapse of prototype when trained naively.

Our contributions are 4-folds: (1) We propose a curvature sampling strategy to identify informative points (and pixels) for sampling, which enables the first joint differentiable-rendering-based pre-training method for fusion perception. (2) Learnable prototypes are utilized to learn a common feature space and an Expectation-Maximization approach is proposed to train the prototypes to represent parts of the 3D scene. (3) We further propose to use a swapping prediction loss for modality interplay and a Gram Matrix Regularization loss that avoids collapse of prototype learning. (4) Through extensive experiments on the popular autonomous driving datasets NuScenes (Caesar et al., 2020) and Waymo (Sun et al., 2020), we demonstrate the effectiveness of CLAP . For example, CLAP achieves up to $100\%$ more improvement than previous SOTA 3D pre-training methods and shows potential scaling property.

## 2 RELATED WORK

**Fusion 3D Object Detection.** Light-Detection-And-Ranging (LiDAR) and camera are important sensors for autonomous driving perception. Previous works mainly focus on single-modality 3D perception. For LiDAR-based 3D object detection, there are three main streams with different em-

bedding schemes for point clouds inputs. 1) Point-based methods (Shi et al., 2019; 2020b) utilize point-level embeddings for 3D object detection. 2) Voxel-based methods (Yan et al., 2018; Deng et al., 2021; Yin et al., 2021; Bai et al., 2022; Yang et al., 2018; Fan et al., 2021) voxelize the 3D scene and use sparse convolution or transformer for embedding. 3) Point-voxel-combined methods (Shi et al., 2020a; 2021) utilize both embeddings from 1) and 2). For camera-based 3D perception, (Ding et al., 2020; Chen et al., 2016; Simonelli et al., 2019; Wang et al., 2021b; Zhang et al., 2021; Ku et al., 2019) embed 2D features on image plane and project these 2D features into 3D space with estimated depth. Recently, research starts to focus on fusion 3D perception (Chen et al., 2017; Liu et al., 2023; Liang et al., 2022; Li et al., 2022b; Liang et al., 2019; Li et al., 2022a; Meyer et al., 2019) with camera images and LiDAR point clouds as inputs. These methods focus on integrating embeddings of different modalities and train in an supervised manner. There are other works on fusion-based semantic segmentation (Wu et al., 2024) and object tracking (Liu et al., 2024). As labeling in 3D space is costly, we explore unsupervised 3D pre-training for fusion 3D perception.

**3D Pre-training.** Annotating for 3D data is notoriously time- and energy-consuming and the emergence of unsupervised representation learning for 2D image (He et al., 2020; Tian et al., 2019; Caron et al., 2020; Grill et al., 2020; Wang et al., 2021c; He et al., 2022) provides a promising way to alleviate the annotation burden. Existing works (et al, 2020; Liu et al., 2020; Hou et al., 2021; Chen et al., 2022; Huang et al., 2021; Liang et al., 2021; Liu et al., 2021a; Sautier et al., 2022; Pang et al., 2023; Yang et al., 2023; et al, 2023; Huang et al., 2023; Zhu et al., 2023) in unsupervised 3D representation learning into scene-level 3D point clouds can be divided into two contrastive-based and masked-and-reconstruction-based paradigms. Contrastive-based works (et al, 2020; Liu et al., 2020; Hou et al., 2021; Chen et al., 2022; Huang et al., 2021; Liang et al., 2021; Pang et al., 2023; Liu et al., 2021a) propose various ways to build suitable views and conduct contrastive learning to improve the performance in downstream perception task. Inspired by (He et al., 2022) in image domain, (Yang et al., 2023; et al, 2023; Huang et al., 2023; Zhu et al., 2023) propose to first mask the input point clouds and pre-train the 3D encoders with a shallow decoder for reconstructing the unmasked inputs. (Sun et al., 2023; Zhang et al., 2024; Li et al., 2022c) are pioneering works to introduce contrastive learning into fusion perception. They consider camera and LiDAR embeddings as different views of the scene and apply the contrastive loss between the two modalities. Inspired by the success of neural field in representing 3D scenes (Mildenhall et al., 2021; Wang et al., 2021a) and previous attempts to introduce neural rendering to 3D pre-training for point clouds (Yan et al., 2023; Huang et al., 2023; Zhu et al., 2023), UniPAD (Yang et al., 2024) proposes to use a differentiable-rendering decoder for masked-and-reconstruction pre-training and achieves SOTA performance for unsupervised 3D representation learning on fusion 3D perception. However, due to the high GPU memory consumption, UniPAD (Yang et al., 2024) is only able to separately pre-train the image and point cloud encoders and fails to utilize the interaction between modalities during pre-training. In this paper, we explore unsupervised joint pre-training for 2D and 3D backbones via differentiable rendering with Curvature Sampling and Prototype Learning. Previous work like SwAV (Caron et al., 2020) is related to our prototype learning part. SwAV (Caron et al., 2020) explores learnable prototypes to represent different categories in the same modality (image) and uses swapping assignment prediction loss for different views of the same image instance. On the contrary, learnable prototypes in CLAP are used to represent part of the 3D scenes and learn the interaction between modalities, which differs from the context in (Caron et al., 2020). Thus we propose a Expectation-Maximization training scheme to maximize similarity between prototypes and 3D embeddings. Then a swapping prediction loss is used to learn the modality interaction. Furthermore, to avoid prototypes collapsing to the same vector, we propose a Gram Matrix Regularization loss.

## 3 METHOD

In this section, we introduce CLAP for joint unsupervised 3D pre-training via differentiable rendering on fusion 3D perception. As described in Fig. 2, CLAP pre-trains the image, LiDAR and fusion encoders jointly with Neural Field Rendering. In order to enable joint pre-training, Curvature Sampling is proposed as shown in (a) of Fig. 2. To further make use of both modalities, we utilize learnable prototypes to represent parts of the 3D scenes as shown in Fig. 2 (b). To optimize the learnable prototype, we train a common feature space with an Expectation-Maximization training scheme and incorporate interaction between modalities by a swapping prediction loss. Finally a Gram Matrix Regularization loss is proposed to avoid collapse in prototype learning. We first discuss the formulation and overall pipeline in Section 3.1. Then we introduce the details about neural

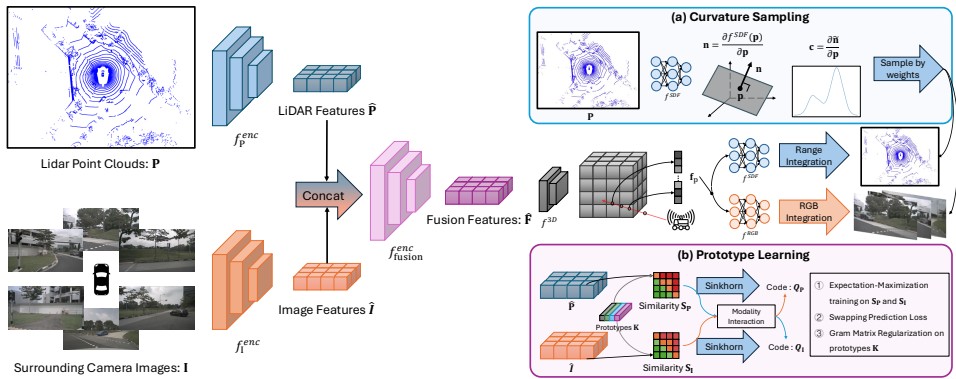

Figure 2: The pipeline of CLAP. In order to jointly pre-train the LiDAR, camera and fusion encoders, we first embed the paired LiDAR point clouds and camera images with $f_{\mathrm{P}}^{\mathrm{enc}}$, $f_{\mathrm{I}}^{\mathrm{enc}}$ and $f_{\mathrm{fusion}}^{\mathrm{enc}}$. Then based on the fusion features, CLAP applies differentiable rendering to predict both range and rgb with the SDF and RGB values of the sampled points along LiDAR/camera rays from $f^{\mathrm{SDF}}$ and $f^{\mathrm{RGB}}$, with which we compute loss against the observed LiDAR point cloud and camera images. To make joint pre-training feasible, we propose Curvature Sampling to sample informative parts of the 3D scene, as described in (a). Furthermore, we propose to use learnable prototypes to represent parts of objects in a common feature space and utilize an Expectation-Maximization approach to maximize the similarity between prototypes and 3D embeddings of each modality. To delve deeper into the interplay of image semantics and LiDAR geometry, we use swapping prototype prediction loss. Finally, we propose a Gram Matrix Regularization loss to prevent collapse of prototype learning.

field and differentiable rendering in Section 3.2. Finally, we describe the curvature sampling and prototype learning separately in Section 3.3 and 3.4.

## 3.1 FORMULATION AND PIPELINE

**Notations.** To begin with, we denote the input image set from $N_{\mathrm{cam}}$ cameras as $\mathcal{I} = \{\mathbf{I}_n \in \mathbb{R}^{H \times W \times 3}\}_{n=1}^{N_{\mathrm{cam}}}$ and LiDAR point cloud as $\mathbf{P} \in \mathbb{R}^{N_p \times (3+d)}$. $H$ and $W$ are the height and width of the images and each pixel on the images has 3 values for RGB. $N_p$ is the number of points in the LiDAR point cloud and each of them contains $xyz$-location and $d$ feature channels. For example, in NuScenes (Caesar et al., 2020) dataset, $d = 2$ represents the intensity and timestamp of each point and there are $N_{\mathrm{cam}} = 6$ surrounding cameras on the autonomous vehicle. For each pair of camera image and LiDAR point cloud, we have the transformation matrix $\mathbf{T}_n \in \mathbb{R}^{3 \times 4}$ indicating the projection between the camera plane and LiDAR coordinate, where $n = 1, 2, ..., N_{\mathrm{cam}}$.

**Encoding.** The goal of unsupervised 3D representation learning for fusion perception is to pre-train the LiDAR, camera and fusion encoder in an unsupervised manner. Hence, we first voxelize and embed the raw LiDAR point cloud $\mathbf{P}$ with LiDAR encoder $f_{\mathrm{P}}^{\mathrm{enc}}$

$$\hat{\mathbf{P}} = f_{\mathrm{P}}^{\mathrm{enc}}(\mathbf{P}), \qquad (1)$$

where $\hat{\mathbf{P}} \in \mathbb{R}^{\hat{D} \times \hat{H} \times \hat{W} \times \hat{d}_{\mathrm{P}}}$ is the embedded 3D features for LiDAR point cloud. $\hat{D}$, $\hat{H}$ and $\hat{W}$ are spatial resolutions of the embedded features and $\hat{d}_{\mathrm{P}}$ is number of feature channels after encoding. Then for camera images $\mathcal{I}$, $f_{\mathrm{I}}^{\mathrm{enc}}$ encodes them with swin transformer (Liu et al., 2021b) and uses $\mathcal{T} = \{\mathbf{T}_n\}_{n=1}^{N_{\mathrm{cam}}}$ to project the 2D features to 3D space.

$$\hat{\mathbf{I}} = f_{\mathrm{I}}^{\mathrm{enc}}(\mathcal{I}, \mathcal{T}, \mathbf{P}), \qquad (2)$$

where $\hat{\mathbf{I}} \in \mathbb{R}^{\hat{D} \times \hat{H} \times \hat{W} \times \hat{d}_{\mathrm{I}}}$ is the embedded 3D features for surrounding camera images with the similar dimensions as $\hat{\mathbf{P}}$ except for $\hat{d}_{\mathrm{I}}$ feature channels. The projection of 2D features to 3D space is similar to (Liu et al., 2023): we transform LiDAR points back to image planes with $\mathcal{T}$ and use the projected ranges from LiDAR to project the 2D features to 3D space. With $\hat{\mathbf{P}}$ and $\hat{\mathbf{I}}$, we further concatenate them along feature dimension and apply the fusion encoder $f_{\mathrm{fusion}}^{\mathrm{enc}}$ to get the fusion

feature $\hat{\mathbf{F}} \in \mathbb{R}^{\hat{D} \times \hat{H} \times \hat{W} \times \hat{d}_\text{F}}$ with $\hat{d}_\text{F}$ feature dimensions,

$$\hat{\mathbf{F}} = f_\text{fusion}^\text{enc}([\mathbf{P}, \mathbf{I}]). \tag{3}$$

**Loss Function.** To guide $f_\text{P}^\text{enc}$, $f_\text{I}^\text{enc}$ and $f_\text{fusion}^\text{enc}$ to learn good representations in an unsupervised manner, CLAP first embed the fusion features $\hat{\mathbf{F}}$ with a shallow 3D convolution network $f^\text{3D}$ to get $\tilde{\mathbf{F}} = f^\text{3D}(\hat{\mathbf{F}})$ and we have $\tilde{\mathbf{F}} \in \mathbb{R}^{\hat{D} \times \hat{H} \times \hat{W} \times \hat{d}_\text{F}}$. Then a rendering loss $\mathcal{L}_\text{rend}$ is applied on $\tilde{\mathbf{F}}$ for masked-reconstruction on both point clouds $\mathbf{P}$ and images $\mathcal{I}$. Furthermore, a prototype learning scheme $\mathcal{L}_\text{proto}$ is utilized in order to bridge the two modalities and incorporate interaction of image semantics and LiDAR geometry into pre-training. The overall loss function is as below:

$$\mathcal{L} = \omega_\text{r} \times \mathcal{L}_\text{rend}(\mathbf{P}, \tilde{\mathbf{F}}, \mathcal{I}) + \omega_\text{proto} \times \mathcal{L}_\text{proto}(\hat{\mathbf{P}}, \hat{\mathbf{I}}), \tag{4}$$

with $\omega_\text{r}$ and $\omega_\text{proto}$ as weighting parameters to balance the losses.

## 3.2 Neural Field and Differentiable Rendering

Inspired by the success of UniPAD (Yang et al., 2024), CLAP applies a differentiable rendering decoder with neural field to conduct the masked-and-reconstruction pre-training. Different from (Yang et al., 2024), an additional surface signed distance field loss is applied to better optimize the scene geometry. Here we first introduce Neural Field, which is the basis for camera images and point clouds rendering, and then discuss the differentiable rendering process on range and RGB values.

**Neural Field.** Given a specific point $\mathbf{p} = [x, y, z] \in \mathbb{R}^3$ in the 3D space, the feature $\mathbf{f}_\text{p} \in \mathbb{R}^{\hat{d}_\text{F}}$ at $\mathbf{p}$ is queried from the fusion 3D embedding $\tilde{\mathbf{F}}$ by trilinear interpolation

$$\mathbf{f}_\text{p} = f^\text{tri}(\mathbf{p}, \tilde{\mathbf{F}}), \tag{5}$$

where $f^\text{tri}$ is an built-in module implemented in Pytorch (Paszke et al., 2019). Taking the concatenation of location $\mathbf{p}$ and queried feature $\mathbf{f}_\text{p}$ as inputs, we predict the signed distance value $s \in \mathbb{R}$ (Chan & Zhu, 2005; Malladi et al., 1995) and color value $c \in \mathbb{R}^3$ (Wang et al., 2021a) at $\mathbf{p}$ with $f^\text{SDF}$ and $f^\text{RGB}$. $f^\text{RGB}$ and $f^\text{SDF}$ are parameterized by Multi-layer Perceptron.

$$s = f^\text{SDF}([\mathbf{p}, \mathbf{f}]) \ , \ c = f^\text{RGB}([\mathbf{p}, \mathbf{f}]), \tag{6}$$

**Differentiable Rendering.** Similar to (Mildenhall et al., 2021; Wang et al., 2021a), we first sample $N_\text{L}$ or $N_\text{C}$ rays at the LiDAR or camera sensor origin $\mathbf{o}$, each of which is described by its normalized direction $\mathbf{d}$ and $\mathbf{o}$. Next, we sample $N_\text{ray}$ points following (Wang et al., 2021a) along each ray. Here each point along the ray can be interpreted by $\mathbf{p} = \mathbf{o} + r\mathbf{d}$, where $r$ is the range from the sensor origin to the point $\mathbf{p}$. Thus the sampled point set can be annotated by $\{\mathbf{p}_n = \mathbf{o} + r_n\mathbf{d}\}_{n=1}^{N_\text{ray}}$ and we predict the estimated signed distance value $s_n$ and color value $c_n$ for them with $f^\text{RGB}$ and $f^\text{SDF}$. Following (Wang et al., 2021a), we estimate the occupancy value $\alpha_n$ for each sampled point,

$$\alpha_n = \max\left(\frac{\Phi_h(s_n) - \Phi_h(s_{n+1})}{\Phi_h(s_n)}, 0\right). \tag{7}$$

Here $\Phi_h(x) = (1 + e^{-hx})^{-1}$ stands for the sigmoid function paired with a learnable scalar $h$. After that, we predict the accumulated transmittance $t_n$ similar to (Wang et al., 2021a)

$$t_n = \prod_{i=1}^{n-1} (1 - \alpha_i). \tag{8}$$

Based on $t_n$, we compute an unbiased and occlusion-aware weight $w_n = t_n \alpha_n$ (Wang et al., 2021a) and integrate all samples along the ray to predict the range $\tilde{r}$ or color $\tilde{c}$ along this ray,

$$\tilde{r} = \sum_{n=1}^{N_\text{ray}} w_n * r_n \ , \ \tilde{c} = \sum_{n=1}^{N_\text{ray}} w_n * c_n \tag{9}$$

For the observed LiDAR points, it is evident that signed distance value should be 0. The loss function for differentiable rendering is a combination of L-1 loss on surface SDF, range and color predictions.

$$\mathcal{L}_\text{rend} = \frac{1}{N_\text{L}} \sum_{i=1}^{N_\text{L}} (|r_i - \tilde{r}_i| + \omega_\text{sur}|s_i|) + \frac{\omega_\text{C}}{3 \cdot N_\text{C}} \sum_{i=1}^{N_\text{C}} \sum_{j=1}^{3} |c_i^j - \tilde{c}_i^j|, \tag{10}$$

where $\omega_{\text{sur}}$ and $\omega_{\text{C}}$ are weighting parameters for the losses. $r_i$ is the observed range along the $i^{th}$ sampled ray and $s_i$ is the predicted signed distance value at the observed points. $c_i^j$ is the value of $j^{th}$ channel in the image pixel, where $j = \{1, 2, 3\}$ corresponds to RGB channels.

### 3.3 CURVATURE SAMPLING

In order to make joint unsupervised representation learning feasible, we have to make $N_{\text{L}} \ll N_{\text{P}}$ and $N_{\text{C}} \ll H \cdot W \cdot N_{\text{cam}}$. Intuitively, uniform sampling with range can be used, same as "Memory-friendly Ray Sampling" in (Yang et al., 2024). But due to the relatively small sample number compared to the raw inputs ($\sim \frac{1}{100}$), this sampling method brings little improvement against separate pre-training, which is contradictory to our motivation. Hence, we need to sample more informative part of the scene for $\mathcal{L}_{\text{rend}}$. As shown in Figure 3, we are inspired by the observation that surface with higher curvature (surface of a vehicle) generally contains more information as compared to that with lower curvature (road plane). Therefore, we propose the Curvature Sampling for effective sampling. In order to make it more intuitive, we show camera image here and the curvature estimation is actually computed in 3D with the SDF function. For each point $\mathbf{p}$ in the LiDAR point cloud $\mathbf{P}$, we first estimate surface normal by deriving the signed distance function with respect to $\mathbf{p}$

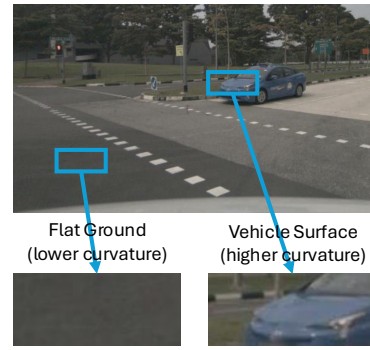

Figure 3: Inspiration of Curvature Sampling.

$$\mathbf{n} = \frac{\delta f^{\text{SDF}}([\mathbf{p}, \mathbf{f}])}{\delta \mathbf{p}}, \tag{11}$$

where $\mathbf{n} \in \mathbb{R}^3$ is the predicted normal. Then we normalized $\mathbf{n}$ to get the direction of the normal $\tilde{\mathbf{n}} = \frac{\mathbf{n}}{||\mathbf{n}||_2}$. Here $|| \cdot ||_2$ is the L-2 norm. Next we apply differential operator on $\tilde{\mathbf{n}}$ with respect to $\mathbf{p}$

$$\mathbf{c} = \frac{\delta \tilde{\mathbf{n}}}{\delta \mathbf{p}}. \tag{12}$$

and get $\mathbf{c} \in \mathbb{R}^3$. Then we estimate the geodesic curvature (Toponogov, 2006) of each point $\mathbf{p}_n$ in LiDAR point cloud by computing the norm of $\mathbf{c}_n$ and use it as the sampling weights $\omega_n$, that is $\omega_n = ||\mathbf{c}_n||_2$. With $\omega_n$, we sample $N_{\text{L}}$ points with a Multinomial Sampler implemented in PyTorch (Paszke et al., 2019) for differentiable rendering. For pixels on image plane, we project the LiDAR point cloud back to image planes with $\mathcal{T}$, assign $\omega_n$ of each point to the projected pixel and apply a gaussian blur kernel of size $K_{\text{gaus}}$ to densify the weights, with which we sample $N_{\text{C}}$ pixels for $\mathcal{L}_{\text{rend}}$. As the curvature estimation is noisy especially in the first few epochs, we apply uniform sampling to warm-up for $N_{\text{warmup}}$ epochs and after that Curvature Sampling is utilized. We implement curvature estimation within torch.no_grad() context, so that it is computed only once and not stored. Thus the *computational and GPU memory overhead* is less than 1% and negligible.

### 3.4 PROTOTYPE LEARNING

Curvature Sampling allows for joint pre-training, prompting us to explore how camera and LiDAR data can be used to understand "objectness" or "object parts" in an unsupervised way. We use learnable prototypes to represent segments of 3D scenes and establish a shared feature space that connects the two modalities. Firstly, we randomly initialize $N_{\text{K}}$ learnable prototypes $\mathbf{K} \in \mathbb{R}^{N_{\text{K}} \times d_{\text{K}}}$, each of which is a $d_{\text{K}}$ vectors. Then an Expectation-Maximization training approach is proposed to maximize the similarity between these prototypes and 3D embeddings from the two modalities. To delve deeper into the interplay between image semantics and LiDAR geometry, we use the swapping prototype prediction loss. Finally, to avoid the prototype collapsing into one same vector, we introduce a Gram Matrix Regularization Loss.

**Expectation-Maximization.** In order to guide the learnable prototypes to represent parts of the environment, we propose an Expectation-Maximization (Moon, 1996) training scheme to optimize

the prototypes. We first project the LiDAR embeddings $\hat{\mathbf{P}}$ and camera embeddings $\hat{\mathbf{I}}$ separately with two projection heads $f_\mathrm{P}^\mathrm{proj}$ and $f_\mathrm{I}^\mathrm{proj}$ to the same dimension as $\mathbf{K}$

$$\dot{\mathbf{P}} = f_\mathrm{P}^\mathrm{proj}(\hat{\mathbf{P}}) \ , \ \dot{\mathbf{I}} = f_\mathrm{I}^\mathrm{proj}(\hat{\mathbf{I}}), \tag{13}$$

where $f_\mathrm{P}^\mathrm{proj}$ and $f_\mathrm{I}^\mathrm{proj}$ are parameterized by Multi-Layer Perceptron and $\dot{\mathbf{P}} \in \mathbb{R}^{\hat{D} \times \hat{H} \times \hat{W} \times d_\mathrm{K}}$, $\dot{\mathbf{I}} \in \mathbb{R}^{\hat{D} \times \hat{H} \times \hat{W} \times d_\mathrm{K}}$. We denote $N_\mathrm{3D} = \hat{D} \times \hat{H} \times \hat{W}$ and then normalize and reshape the projected embeddings into $\dot{\mathbf{P}} \in \mathbb{R}^{N_\mathrm{3D} \times d_\mathrm{K}}$ and $\dot{\mathbf{I}} \in \mathbb{R}^{N_\mathrm{3D} \times d_\mathrm{K}}$. After that, similarity scores $\mathbf{S}_\mathrm{P/I} \in \mathbb{R}^{N_\mathrm{3D} \times N_\mathrm{K}}$ between 3D embeddings and prototypes are computed separately for LiDAR and camera branches

$$\mathbf{S}_\mathrm{P} = \dot{\mathbf{P}} \cdot \mathbf{K}^\mathsf{T} \ , \ \mathbf{S}_\mathrm{I} = \dot{\mathbf{I}} \cdot \mathbf{K}^\mathsf{T}. \tag{14}$$

In the Expectation step, we compute the probability $\hat{\mathbf{S}}_\mathrm{P/I}$ that each prototype is assigned to each embeddings by applying a softmax operation on $\mathbf{S}_\mathrm{P/I}$. Then for Maximization step, we expect to maximize the probability of the assignment between one prototype to one specific part of the scene and this is equal to minimize the entropy of the similarity matrix. Thus, the EM loss is computed as

$$\mathcal{L}_\mathrm{EM} = -\frac{1}{N_\mathrm{3D} N_\mathrm{K}} \sum_{n=1}^{N_\mathrm{3D}} \sum_{m=1}^{N_\mathrm{K}} \{\hat{\mathbf{S}}_\mathrm{P}^{n,m} log \hat{\mathbf{S}}_\mathrm{P}^{n,m} + \hat{\mathbf{S}}_\mathrm{I}^{n,m} log \hat{\mathbf{S}}_\mathrm{I}^{n,m}\}. \tag{15}$$

**Swapping Prototype Prediction.** To further explore interaction between modalities, we detach $\mathbf{S}_\mathrm{P/I}$ and apply sinkhorn algorithm (Cuturi, 2013) to approximate them to double stochastic matrix in $N_\mathrm{sink}$ iterations. We denote the updated matrix as codes $\mathbf{Q}_\mathrm{P/I} \in \mathbb{R}^{N_\mathrm{3D} \times N_\mathrm{K}}$. The swapping prototype prediction loss is computed with a temperature parameter $\tau$, inspired by (Caron et al., 2020),

$$\begin{aligned} \mathcal{L}_\mathrm{SwAV} = -\frac{1}{N_\mathrm{3D} N_\mathrm{K}} \sum_{n=1}^{N_\mathrm{3D}} \sum_{m=1}^{N_\mathrm{K}} \{\mathbf{Q}_\mathrm{I}^{n,m} \log \frac{exp(\mathbf{S}_\mathrm{P}^{n,m})/\tau}{\sum_{k=1}^{N_\mathrm{K}} exp(\mathbf{S}_\mathrm{P}^{n,k})/\tau} \\ + \mathbf{Q}_\mathrm{P}^{n,m} \log \frac{exp(\mathbf{S}_\mathrm{I}^{n,m})/\tau}{\sum_{k=1}^{N_\mathrm{K}} exp(\mathbf{S}_\mathrm{I}^{n,k})/\tau}\}. \end{aligned} \tag{16}$$

**Gram Matrix Minimization.** When training the randomly initialized prototypes, the network might learn a short cut with all prototypes being the same (Caron et al., 2020), which is called collapse. To avoid this, we estimate similarity between prototypes by the gram matrix $\mathbf{G} = \mathbf{K}\mathbf{K}^\mathsf{T}$ of prototypes $\mathbf{K}$, the dimension of which is $\mathbf{G} \in \mathbb{R}^{N_\mathrm{K} \times N_\mathrm{K}}$. Finally we minimize the average of the non-diagonal elements of $\mathbf{G}$ in order to avoid collapse

$$\mathcal{L}_\mathrm{GMM} = \frac{1}{N_\mathrm{K}(N_\mathrm{K} - 1)} \sum_{n} \sum_{m=1, m \neq n}^{N_\mathrm{K}} \mathbf{G}^{n,m}. \tag{17}$$

**Overall Prototype Learning Loss.** We apply weighting parameters $\omega_\mathrm{SwAV}$, $\omega_\mathrm{EM}$ and $\omega_\mathrm{GMM}$ to balance the three losses proposed above, which leads to the overall loss function for prototype learning,

$$\mathcal{L}_\mathrm{proto} = \omega_\mathrm{SwAV} \mathcal{L}_\mathrm{SwAV} + \omega_\mathrm{EM} \mathcal{L}_\mathrm{EM} + \omega_\mathrm{GMM} \mathcal{L}_\mathrm{GMM}. \tag{18}$$

## 4 EXPERIMENTS

Unsupervised 3D representation learning for fusion perception aims to pre-train both LiDAR and camera encoders and initialize downstream models with the pre-trained weights to gain performance improvement in downstream tasks. In this section, we design extensive experiments on the popular autonomous driving dataset NuScenes (Caesar et al., 2020) and Waymo (Sun et al., 2020) to demonstrate the effectiveness of CLAP. To begin with, we describe experiment setups in Section 4.1. Next, we show and analyze main results in Section 4.2. Finally, we provide ablation study and visualizations separately in Section 4.3 and 4.5.

Table 1: Results for fine-tuning on 5% of training set in NuScenes. "Init." means the way to initialize models. We guarantee convergence of from-scratch model and fix the training iteration for all fine-tuning experiments. We provide mAP and NDS as an evaluation of the overall performance of different models and highlight the best mAP and NDS with bold font. We also indicate the performance improvement by green color. "C.V.", "Mot.", "Bic.", "Ped." and "T.C." are abbreviations for Construction Vehicle, Motorcycle, Bicycle, Pedestrian and Traffic Cone. Results are in %.

| Init. | mAP | NDS | Car | Truck | C.V. | Bus | Trailer | Barrier | Mot. | Bic. | Ped. | T.C. |
|---|---|---|---|---|---|---|---|---|---|---|---|---|
| Rand. | 48.69 | 55.28 | 78.52 | 46.64 | 16.18 | 50.44 | 22.11 | 57.00 | 46.87 | 30.56 | 76.16 | 62.40 |
| ALSO | $45.34^{-3.35}$ | $49.19^{-6.09}$ | 77.61 | 41.98 | 14.61 | 36.09 | 12.55 | 58.10 | 42.89 | 30.56 | 74.09 | 64.98 |
| OCC-MAE | $47.39^{-1.30}$ | $54.97^{-0.31}$ | 77.43 | 46.25 | 15.26 | 50.06 | 19.35 | 55.35 | 43.24 | 30.92 | 74.69 | 61.34 |
| SLidR | $47.23^{-1.46}$ | $52.77^{-2.51}$ | 77.12 | 45.04 | 16.19 | 50.50 | 22.17 | 57.74 | 41.47 | 30.22 | 71.11 | 60.71 |
| PPKT | $49.58^{+0.89}$ | $55.85^{+0.57}$ | 79.24 | 47.26 | 17.26 | 51.37 | 21.14 | 59.55 | 44.82 | 31.43 | 78.03 | 65.73 |
| UniPAD | $49.81^{+1.12}$ | $55.29^{+0.01}$ | 80.81 | 42.81 | 17.08 | 48.98 | 25.85 | 61.72 | 50.19 | 27.53 | 78.53 | 64.57 |
| CLAP | $\mathbf{51.17}^{+2.48}$ | $\mathbf{57.04}^{+1.76}$ | 79.56 | 48.43 | 18.84 | 56.34 | 23.98 | 60.60 | 48.87 | 34.11 | 78.08 | 62.87 |

## 4.1 SETTINGS

**Datasets.** We use the popular autonomous driving dataset NuScenes (Caesar et al., 2020) and Waymo (Sun et al., 2020) to evaluate the performance of CLAP. NuScenes (Caesar et al., 2020) uses one roof LiDAR and six surrounding cameras to collect data. The LiDAR is a 32-beam Velodyne and collecting frequency is 20Hz. The frequency of camera capturing is 12Hz. (Caesar et al., 2020) conducts the synchronization and provides paired data of LiDAR point cloud and camera images. The whole NuScenes dataset contains 1000 scenes collected in Boston and Singapore. Each scene lasts for around 20 seconds and there are a total of 5.5 hours data. Following convention practice from (Caesar et al., 2020; Team, 2020), we divide the whole dataset into training set with 850 scenes and validation set with 150 scenes. Waymo (Sun et al., 2020) uses one top 64-beam LiDAR, 4 corner LiDARs and 5 surrounding cameras to collect point clouds and camera images. Following the same practice in (Sun et al., 2020; Team, 2020), the collected 1000 scenes are split into training set (798 scenes) and validation set (202 scenes). All pre-trainings are conducted on the training set without labels and we conduct downstream 3D object detection training in few-shot setting via uniform sampling of the training data (NuScenes 5% and Waymo 1%).

**Downstream 3D Object Detectors.** For NuScenes, we select the SOTA 3D object detector called BEVFusion (Liu et al., 2023) and for Waymo, we use CenterPoint (Yin et al., 2021). Both BEVFusion and CenterPoint are implemented in the popular code repository for autonomous driving perception called OpenPCDet (Team, 2020). For evaluation metrics, we use average precisions of various categories (APs), mean average precision (mAP) and NuScenes Detection Score (NDS) (Caesar et al., 2020) for NuScenes and mAP (mean accurate precisions) and mAPH (mean accurate precisions with headings) at different difficulty levels (Level-1 and 2) for Waymo (Sun et al., 2020). We follow a similar setting in (He et al., 2019; et al, 2023) to gradually increase training iterations of the from-scratch model until convergence is observed. Here convergence means further increasing training iterations will not improve the performance. Then the number of training iterations is fixed for fine-tuning pre-trained models. This setting *avoids the case that pre-training only accelerates convergence and makes sure that pre-training indeed improve the performance of downstream models*, that is improving the sample efficiency of the downstream task.

**Baseline Pre-training Method for Fusion Perception.** We incorporate three kinds of pre-training baseline methods: 1) an occupancy estimation method called ALSO (Boulch et al., 2023), 2) occupancy masked autoencoder called Occupancy-MAE (Min et al., 2023), 3) multi-modality methods including SLidR (Sautier et al., 2022), PPKT (Liu et al., 2021a) and UniPAD (Yang et al., 2024). We use the official implementations to pre-train the backbones with same setting as CLAP.

**Implementation Details of CLAP.** The feature channels for embeddings $\hat{\mathbf{P}}$, $\hat{\mathbf{I}}$, $\hat{\mathbf{F}}$ and prototypes $\mathbf{K}$ are respectively set to $\hat{d}_P = 256$, $\hat{d}_I = 80$, $\hat{d}_F = 512$ and $\hat{d}_K = 128$. Sampling number for point cloud and pixel are $N_L = 8192$ and $N_C = 1024 \times N_{cam}$. The number of sample points along each ray is $N_{ray} = 96$. Warm-up epochs for Curvature Sampling is $N_{warmup} = 4$. We set the number of learnable prototypes and sinkhorn update iterations to $N_K = 512$ and $N_{sink} = 3$. The temperature for swapping prediction loss is $\tau = 1.0$. The loss weighting parameters are implemented as $\omega_r = 2.0$, $\omega_{proto} = 1.0$, $\omega_{sur} = 0.05$, $\omega_C = 0.05$, $\omega_{SwAV} = 1.0$, $\omega_{EM} = 0.1$ and $\omega_{GMM} = 0.1$. We use torch.auto_grad() (Paszke et al., 2019) to implement the derivatives in the curvature estimation. More details about pre-training and fine-tuning are provided in Appendix A.

Table 2: Results for Waymo (Sun et al., 2020).

| Init. | Level-1 | | Level-2 | | $\bar{\Delta}$ |
|---|---|---|---|---|---|
| | mAP | mAPH | mAP | mAPH | |
| Rand. | 61.60 | 58.58 | 55.62 | 52.87 | 0 |
| ALSO | 62.09 | 59.03 | 56.12 | 53.32 | +0.47 |
| OCC-MAE | 62.33 | 59.32 | 56.36 | 53.63 | +0.74 |
| SLidR | 62.10 | 59.09 | 56.10 | 53.36 | +0.49 |
| PPKT | 62.32 | 59.22 | 56.37 | 53.55 | +0.69 |
| UniPAD | 61.57 | 58.64 | 55.64 | 52.93 | +0.02 |
| CLAP | **62.87** | **59.88** | **56.88** | **54.16** | **+1.28** |

Table 3: Results on scaling property.

| Init. | F.T. Data | mAP | NDS |
|---|---|---|---|
| Random | 5% | 48.69 | 55.28 |
| CLAP | | 51.17 $^{+2.48}$ | 57.04 $^{+1.76}$ |
| Random | 2.5% | 39.12 | 40.01 |
| CLAP | | 42.86 $^{+3.74}$ | 42.18 $^{+2.17}$ |
| Random | 1% | 26.22 | 29.82 |
| CLAP | | 30.53 $^{+4.31}$ | 31.87 $^{+2.05}$ |
| Random | 0.5% | 16.49 | 22.61 |
| CLAP | | 23.71 $^{+7.22}$ | 27.32 $^{+4.71}$ |

Table 4: Ablation Study. The second line is joint pre-training with sampling method from UniPAD.

| Joint Pre-train | Cur. Sam. | Proto. Learning | mAP |
|---|---|---|---|
| ✗ | ✗ | ✗ | 49.81 |
| ✓ | ✗ | ✗ | 49.55 |
| ✓ | ✓ | ✗ | 50.81 |
| ✓ | ✓ | ✓ | 51.17 |

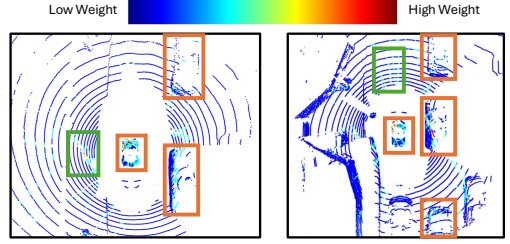

Figure 4: Visualization of curvature estimation.

## 4.2 MAIN RESULTS

**NuScenes Results.** As shown in Table 1, CLAP achieves 2.48% mAP improvement over randomly initialization at convergence, which is $100\%$ more improvement for mAP than SOTA unsupervised 3D representation method UniPAD (Yang et al., 2024) and the best among all initialization methods. For NDS metric, UniPAD (Yang et al., 2024) only achieves comparable performance and PPKT (Liu et al., 2021a) makes a gain of 0.57% while CLAP surpasses the train-from-scratch model by 1.76% . When it turns to different categories, CLAP generally benefit the performance of all the categories and for Construction Vehicle, Bus, Barrier, Motorcycle and Bicycle, the improvement over random initialization are more than 2% AP.

**Waymo Results.** In Table 2, we provide the average performance difference of mAP and mAPH over different difficulty levels. It can be found that CLAP achieves the best performance at convergence. Meanwhile, the performance gain brought by CLAP is approximately two times as the best (OCC-MAE (Min et al., 2023)) of previous pre-training methods. This demonstrates the effectiveness and generalization ability of CLAP.

**Potential Scaling Property.** As we are not able to scale up the pre-training dataset at current stage, we explore potential scaling property by gradually decreasing the sample numbers (2.5%, 1% and 0.5%) for fine-tuning on NuScenes, which increases the ratio between pre-training data and fine-tuning data. The results are shown in Table 3 (F.T. is fine-tuning). It can be found that as the ratio between pre-training data and fine-tuning data gets larger, the performance improvement by CLAP increases and CLAP provides a gain up to 7.22% mAP and 4.71% NDS with 0.5% fine-tuning data. These results show that CLAP is promising in scaling property and in the future, if we can scale up the pre-training dataset, CLAP might further improve current SOTA performance.

## 4.3 VISUALIZATIONS ON CURVATURE ESTIMATION

We use CLAP to estimate the curvature of point clouds and use heatmap color to indicate the weight computed in Section 3.3. As shown in Figure 4 (orange boxes for those regions with relatively correct estimation and green ones for those with noisy estimation), it can be found that though some noise exists (because pre-training is conducted in unsupervised manner), CLAP is able to predict high weights for those highly informative region for sampling and meanwhile assign lower weights to most of the background, which makes joint pre-training feasible.

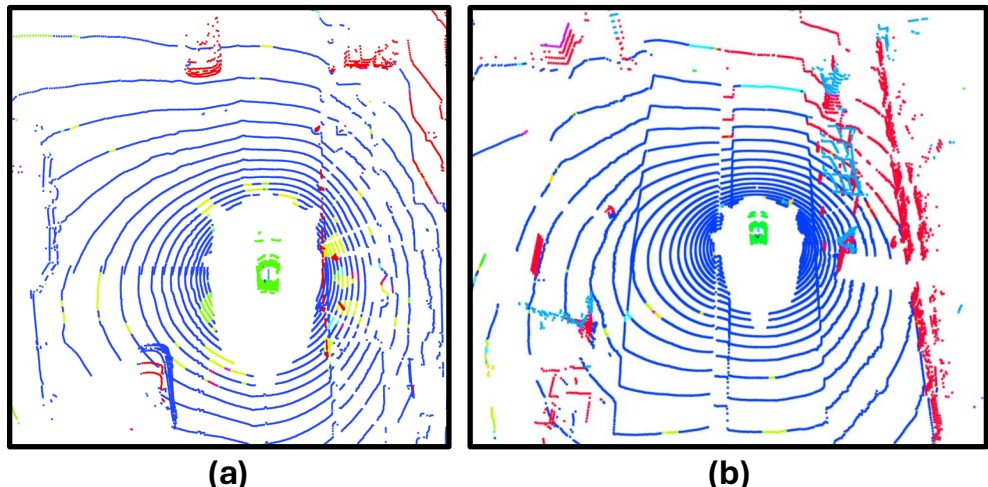

**(a)**         **(b)**

Figure 5: Visualization of the prototype learning results. Different color indicates different prototype assignments.

## 4.4 VISUALIZATIONS ON PROTOTYPE LEARNING

We use the model pre-trained by CLAP to infer the 3D features and assign prototypes to different LiDAR points in the 3D space. Then we use different random colors to indicate different prototypes and visualize them, as shown in Figure 5. If can be found that the background road plane inside the same frame is generally assigned to the same prototype. And foreground vehicles are assigned to another prototype. This demonstrates that the proposed prototype learning scheme actually learns to represent parts of the scenes with prototypes in an unsupervised manner. However, as our pre-training does not incorporate any label, it can also be found that the prototype assignment has some noise, for example some of the road plane points are assigned to other prototypes.

## 4.5 ABLATION STUDY

We conduct ablation study to evaluate the effectiveness of different components. Results are in Table 4. The first line is seperate pre-training with UniPAD (Yang et al., 2024). The second one is jointly pre-training using UniPAD (Yang et al., 2024) and uniform sampling guided by range ("Memory-friendly Ray Sampling" in (Yang et al., 2024)) to address the GPU memory limitation.

It can be found that using simple sampling method does not bring improvement over separate pre-training. Then we add Curvature Sampling in third line and found that Curvature Sampling enables more effective sampling and improves the performance over uniform sampling guided by range and separate pre-training. Finally, the Prototype Learning scheme (fourth line) learns a common feature space for segments of 3D scenes and introduces interaction of LiDAR and camera encoders, which achieves the best performance.

## 5 CONCLUSION

In this paper, we propose CLAP for unsupervised fusion joint pre-training via differentiable rendering. CLAP uses Curvature Sampling to sample more informative parts and learnable prototypes to represent parts of 3D scenes, optimized with an Expectation-Maximization approach. To explore the interplay between LiDAR geometry and image semantics, a swapping prediction loss is used with a regularization loss to avoid collapse. Experiment results demonstrate that CLAP is superior in unsupervised 3D representation learning and has the potential to scale up.

## ACKNOWLEDGMENTS

This paper is supported by the General Research Fund of Hong Kong No.17209324 and 17200622, and the Global Industrial Technology Cooperation Center (GITCC) through a grant agreement with the Korea Institute for Advancement of Technology (KIAT), project number P0028922.

## ETHICS STATEMENT

We further discuss potential negative social impact of CLAP in this section.

*Job Displacement.* Automation of tasks that require 3D perception, such as autonomous vehicles or robotics, might lead to job losses in sectors like transportation, warehousing, and manufacturing. While automation can create new jobs, the transition period can be challenging for those whose jobs are automated away.

*Security Risks.* The deployment of autonomous systems could introduce new vulnerabilities. Any failures in autonomous driving or robotics could lead to accidents or be exploited maliciously.

*Accessibility and Inequality.* The benefits of advanced 3D perception technologies might not be evenly distributed across society. Wealthier regions or organizations may have earlier and better access to these technologies, potentially widening the gap between different socioeconomic groups.

## REPRODUCIBILITY STATEMENT

We provide method details in Section 4 and implementation details in Appendix A, which are enough for reproducing the experiments. Besides, we will release the code and models.

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

## A  MORE IMPLEMENTATION DETAILS

**Pre-training.** We use a learning rate of 0.00005 with a cosine learning schedule for pre-training and use mask augmentation for CLAP with a masking rate of 0.9. All the pre-trainings are conducted on 8-H100 clusters for similar time ($\sim$ 45 mins / epoch). As pre-trained image backbones are broadly used for image feature extraction, the implementation of Yang et al. (2024) also use a pre-trained image backbone from Contributors (2020) to initiate their pre-training and the training set for this backbone does not involve any data from NuScenes Caesar et al. (2020) and Waymo Sun et al. (2020). We adopt this practice for training-from-scratch model and CLAP.

**Downstream Training.** We follow the common practice in OpenPCDet Team (2020) and only change the training iterations in order to observe the convergence of train-from-scratch models, which avoids the case that pre-training only accelerate convergence and make sure that pre-training indeed improve the performance of downstream models. The training epoch is 108 for BEVfusion Liu et al. (2023) with $5\%$ of NuScenes Caesar et al. (2020) training data and 252 for CenterPoint Yin et al. (2021) with $1\%$ of Waymo Sun et al. (2020) training data.

## B  MORE EXPERIMENT RESULTS

### B.1  SEMANTIC SEGMENTATION

We fine-tune Cylinder3D Zhou et al. (2020) for LiDAR semantic segmentation on Semantic KITTI dataset Behley et al. (2021). We use mIoU as eval metric. Results are $28.23\%$ for random initialization, $31.88(+3.55)\%$ for UniPAD and $34.28(+6.05)\%$ for CLAP. It can be found that CLAP is able to benefit different tasks and achieves $70\%$ more improvement than UniPAD Yang et al. (2024).

### B.2  REPEATED EVALUATION

We use the same fixed random seed for all experiments in the main paper for reproducibility. As repeated evaluation can further reveal the training robustness, we repeat fine-tunings on NuScenes for 5 times with random initialization, UniPAD and CLAP. Mean and standard deviation of mAP are $48.55 \pm 0.18\%$ (Rand.), $49.66 \pm 0.29\%$ (UniPAD) and $51.16 \pm 0.10\%$ (CLAP). CLAP achieves the best average performance and robustness against random seeds.

### B.3  TRANSFERRING TO OTHER DATASETS

We conduct further experiments to evaluate the transferring ability of CLAP. Specifically, we select LiDAR-based 3D object detector CenterPoint Yin et al. (2021) on Once Mao et al. (2021) for the downstream task. A 40-beam LiDAR is utilized in Once Mao et al. (2021) to collect 15k labeled training data. We randomly sample $5\%$ and also use all of the labeled training set to train the from-scratch model until convergence is observed. Then we use pre-trained weights by CLAP on NuScenes Caesar et al. (2020) to initialize the same model and fine-tune it with the same training iterations as the randomly initialized model. Results are shown in Table 5. It can be found that pre-training by CLAP also benefits LiDAR-based 3D object detection, even in a cross-dataset setting. And if we look at the performance of "Rand*" and "CLAP*", CLAP also accelerates the convergence in downstream task.

### B.4  MORE FINE-TUNING DATA

We fine-tune on $100\%$ NuScenes and Waymo with fixed random seed to provide more insights. Here, we show mAP and NDS for NuScenes in $2^{nd}$ and $3^{rd}$ cols and **Level-2** mAP, mAPH for Waymo. CLAP improves random initialization by $+0.55$ whereas UniPAD by $+0.15$ on average, showing CLAP's effectiveness. With Table 3 in the paper, more gain can be expected with more unlabeled data in real scenario.

| Init. | F.T. | mAP | Vehicle | | | Pedestrian | | | Cyclist | | |
|-------|------|-----|---------|-------|------|-----------|-------|------|---------|--------|------|
| | | | 0-30m | 30-50m | 50m- | 0-30m | 30-50m | 50m- | 0-30m | 30-50m | 50m- |
| Rand* | | 20.48 | 58.03 | 25.22 | 12.98 | 11.62 | 9.75 | 6.97 | 21.55 | 6.83 | 3.11 |
| CLAP* | 5% | 22.86 $^{+2.38}$ | 58.37 | 26.38 | 14.07 | 12.60 | 9.50 | 7.88 | 30.08 | 10.50 | 5.39 |
| Rand | | 46.07 | 76.71 | 51.15 | 31.84 | 37.53 | 20.12 | 9.84 | 62.00 | 42.61 | 24.18 |
| CLAP | | 46.88 $^{+0.81}$ | 76.98 | 51.64 | 31.31 | 38.79 | 20.60 | 9.74 | 63.75 | 43.21 | 26.83 |
| Rand* | | 64.00 | 86.21 | 70.20 | 58.20 | 57.80 | 41.18 | 23.55 | 75.95 | 61.45 | 45.80 |
| CLAP* | 100% | 64.74 $^{+0.74}$ | 88.14 | 72.59 | 59.13 | 57.37 | 42.24 | 24.22 | 77.11 | 61.91 | 45.63 |
| Rand | | 65.03 | 88.18 | 74.23 | 61.75 | 57.32 | 38.90 | 21.96 | 78.07 | 64.32 | 48.16 |
| CLAP | | 65.56 $^{+0.53}$ | 87.97 | 72.77 | 62.11 | 58.33 | 40.11 | 21.29 | 78.63 | 64.70 | 47.27 |

Table 5: Results for transferring experiments on Once Mao et al. (2021) dataset. CenterPoint Yin et al. (2021) is used as the downstream detector. "Init." indicates the initialization methods. "F.T." indicates the number of training samples in fine-tuning stage. Overall mAP and APs for different categories within different ranges are shown in this table. "Rand*" means training the randomly initialized model with the original training iterations in OpenPCDet Team (2020). "Rand" indicates that we increase the number of training iterations for randomly initialized model until convergence is observed. "CLAP*" indicates that we pre-train the backbones with CLAP on NuScenes Caesar et al. (2020) and then fine-tune on Once with the original iterations in Team (2020). "CLAP" uses the same fine-tuning iterations as "Rand". We use green color to highlight the performance improvement brought by CLAP. All the results are in %.

| Init. | mAP | NDS | mAP | mAPH |
|-------|------|------|------|------|
| Rand. | 64.57 | 67.61 | 69.00 | 67.57 |
| UniPAD | 64.69 | 68.06 | 69.03 | 67.56 |
| CLAP | 65.11 | 68.38 | 69.49 | 68.04 |

Table 6: Fine-tuning on $100\%$ NuScenes and Waymo with fixed random seed.

## C  COMPARISON TO "3D TRANS".

We respectfully clarify that methods in 3DTrans focus on supervised/semi-supervised pre-training, which requires labels during pre-training, and domain adaptation, which conducts supervised training on one dataset and transfers to other datasets. The focus of 3DTrans is fundamentally different problem settings from our work. The baseline methods in our experiments (ALSO, OCC-MAE, SLidR, PPKT, UniPAD) represent the current SOTA in unsupervised 3D pre-training with LiDAR and camera modalities, which is the specific problem we address. Meanwhile, we agree that including supervised/semi-supervised pre-training methods would enrich our analysis. We provide additional experiments using supervised/semi-supervised methods from 3DTrans (ADPT and SPOT).

| Init. | mAP | NDS |
|-------|------|------|
| From-scratch | 48.69 | 55.28 |
| SPOT | 50.98 | 57.20 |
| ADPT | 51.05 | 56.25 |
| CLAP | 51.17 | 57.04 |

## D  DISCUSSION ON PROTOTYPE LEARNING.

Although prototype learning is established in the SSL community, we emphasize that our contribution is not simply applying existing prototype methods, but rather novel adaptations for 3D scene understanding:

(1) Different purpose: Unlike SwAV Caron et al. (2020), where prototypes represent different categories within the same modality (images), our prototypes represent parts of 3D scenes and bridge two different modalities (LiDAR + camera).

(2) Expectation-Maximization training scheme (Section 3.4): We propose maximizing similarity between 3D embeddings and prototypes by minimizing entropy of assignment probabilities - designed specifically for 3D scene part representation, not category clustering.

(3) Gram Matrix Regularization: A novel regularization term to prevent prototype collapse in the 3D multimodal setting.

(4) Cross-modal swapping prediction: While inspired by SwAV, our swapping operates between different modalities (predicting camera prototypes from LiDAR embeddings and vice versa) rather than different views of the same modality.

Empirical validation: Table 4 shows that prototype learning provides improvement beyond curvature sampling alone, and the combination achieves 100% more improvement than SOTA (2.48% vs. 1.12% from UniPAD).

## E    DISCUSSION ON CLAP AND UniPAD.

We would like to clarify that CLAP represents a substantial departure from UniPAD Yang et al. (2024), which employs differentiable rendering for pre-training. While both methods use neural-rendering decoders (a common approach in the field), our decoder differs by incorporating an additional surface signed distance field loss for better scene geometry optimization. The fundamental distinction is that UniPAD performs separate pre-training of each modality, while CLAP enables joint pre-training with Curvature Sampling and introduces cross-modal interactions via Prototype Learning, which are paradigm shift. CLAP's novel contributions (Curvature Sampling, Prototype Learning with EM training, Swapping prediction loss, and Gram Matrix Regularization) have no counterparts in UniPAD and enable capabilities that UniPAD cannot achieve. Our ablation study in Table 4 quantifies this difference.

## F    THE USE OF LARGE LANGUAGE MODELS.

We use Large Language Models to help and aid editing/polishing the paper. In details, we polish the Introduction and Related Work sections with Large Language Models, which mostly focuses on grammar, spelling and word choice.

