# OpenReview forum: "CLAP: Unsupervised 3D Representation Learning for Fusion 3D Perception via Curvature Sampling and Prototype Learning"
_ICLR.cc/2026/Conference — ICLR 2026 Poster_

### Official Review · Reviewer_tkBv · 2025-10-17

**Soundness:** 4
**Presentation:** 4
**Contribution:** 3
**Rating:** 8
**Confidence:** 5

**Summary:**

This paper presents a joint unsupervised differentiable rendering pre-training method for images and point clouds, called CLAP, curvature sampling and learnable prototype. Specifically, CLAP overcomes computational barriers through curvature sampling, thus selecting more informative points/pixels for pre-training. To reveal the complementary performance advantages of both, it is proposed to use learnable prototypes to represent the various parts of the 3D scene in the common feature space and associate the embedding of each modality with the prototype using an expectation-maximizing training scheme. This paper also proposes an exchange prediction loss that explores their interactions through the prototype and uses a Gram matrix regularization term to maintain training stability. Experiments on the NuScenes and Waymo datasets show performance gains of up to 100% for CLAP compared to previous SOTA pre-training methods.

**Strengths:**

The strengths of this paper are as follows:
1. The research direction of this paper has obvious value. The joint multi-modal unsupervised representation learning has a wide range of application scenarios for real-world applications in 3D vision.
2. The research content of this paper is quite challenging, and there is a shortage of work that can effectively solve the limitations in this field.
3. From the perspective of methods and results, this paper is highly complete and the methods are sufficiently original. For example, the curvature sampling, expectation maximization training, and swap prediction loss are proposed for the first time.
4. The performance quality of this paper is excellent. Although there are not many illustrations and result figures in the text due to content limitations, it is easy for readers to understand and recognize.

**Weaknesses:**

The weaknesses of this paper are as follows:
1. Some elements in Figure 1 should be distinguished, including but not limited to: the sub-images corresponding to the point cloud and rendering loss are consistent in content, which seems to be unable to reflect the role of rendering loss. It is recommended to modify the sub-images corresponding to the rendering loss to have information with object supervision; (a) (b) The element repetition is high. It is recommended to reflect the transformation of the point cloud and rendering loss after using Swapping Prototype Assignment and Curvature Sampling.
2. As the pipeline of the method, Figure 2 shows the depth integration and RGB integration, however, this is not explained in the subtitles or the text. In addition, Nuscenes and waymo do not have depth maps. How is the depth integration achieved in this paper?
3. The rendering loss is mentioned many times in the paper, which should refer to the loss of rendering the point cloud as an image. However, the method introduces the projection of points to pixels through a given projection matrix. The rendering process seems redundant?
4. In Formula 1, the author says that $\hat{\mathbf{P}} \in \mathbb{R}^{\hat{D} \times \hat{H} \times \hat{W} \times \hat{d}_{\mathrm{p}}}$ is the generated embedding feature, which is obviously not per-point features. For each superscript, the author should give a clear explanation, and it is recommended to explain it in text and in Figure 2.
5. Formula 4 needs more introduction. How are $L_{rend}$ and $L_{proto}$ implemented, and what type of loss is used?
6. The example of curvature sampling in Figure 3 is unconvincing to me. Because there is no curvature in a two-dimensional image, it is recommended that the author redraw the drawing as described in the text. Operate on LiDAR point cloud, reflect the signed distance function, surface normal, etc.
7. The experiment section of this paper shows that a pre-trained model is used for fine-tuning, but it is not accurate to call this an unsupervised method.
8. Figure 4 obviously needs to be improved. It is recommended that the author add sample sub-images in a row to fill the blanks. In addition, why are there fewer points with high sampling weights and concentrated at the bottom of the car object? According to theoretical analysis, shouldn't the area with large curvature transformation be concentrated above or around the car? The author must conduct sufficient experimental analysis, as I guess this may be a defect caused by the implicit signal function used.
9. The method of this paper is abbreviated as CLAP (**C**urvature samp**L**ing and le**A**rnable **P**rototype). Is it to be as similar as possible to the classic work CLIP? I think it is easier to understand if CSLP appears in the title. Of course, it depends entirely on the author's ideas.
10. The formulas and mathematical symbols in this paper seem to have flaws. It is recommended to give a complete explanation or separate them by pronunciation, for example, $f^{enc}$ can be changed to $f^{encoder}$ or $f^{en}$. Especially in formulas 13 and 14, there is only one dot above the mathematical symbols. It is recommended to change to a more obvious expression, such as a line or superscript.

**Questions:**

As a reviewer, I reviewed this work on an almost per-word-by-word basis. Because of my obvious interest in it, however, given the current presentation quality and experimental results (see **Weaknesses**) of this paper leave much to be desired, I can only give it a neutral rating.

I very much hope that the authors can provide enough evidence to convince me in the limited time available. If the evidence is sufficient, I will not begrudge my score and may highlight it to AC, conversely, I may feel that it needs enough revisions to reject it.

**Details Of Ethics Concerns:**

No Ethics Concerns

---

> ### Author Response · Authors · 2025-11-20
> **Response to Reviewer tkBv. (Part 1)**
>
> Dear Reviewer tkBv,
>
> Thank you for your precious time on the review and your constructive suggestions to improve our manuscript! We appreciate the acknowledgment that the research direction is challenging and has obvious value, the paper is highly complete and the methods are sufficiently original and the paper is easy to understand and follow.
>
> We provide further discussions on your questions as belows:
>
> **Q1: Some elements in Figure 1 should be distinguished, including but not limited to: the sub-images corresponding to the point cloud and rendering loss are consistent in content, which seems to be unable to reflect the role of rendering loss. It is recommended to modify the sub-images corresponding to the rendering loss to have information with object supervision; (a) (b) The element repetition is high. It is recommended to reflect the transformation of the point cloud and rendering loss after using Swapping Prototype Assignment and Curvature Sampling.**
>
> Thanks for pointing it out. Figure 1 intends to illustrate the difference between previous *separate* pre-trainings and the *joint* pre-training scheme of CLAP. And we want to clarify that there is no ``object supervision'' in unsupervised representation learning. Rendering loss is computed with raw point cloud and image observation. Besides, the overall pipeline is that we first use curvature sampling to sample informative rays for rendering, based on which we render image and point clouds for rendering loss. And Swapping Prototype Assignment is a parallel process to rendering. We will update the figure accordingly in the final version.
>
> **Q2: As the pipeline of the method, Figure 2 shows the depth integration and RGB integration, however, this is not explained in the subtitles or the text. In addition, Nuscenes and waymo do not have depth maps. How is the depth integration achieved in this paper?**
>
> In Section 3.2 of the main paper, line 239 to 257, we discuss the process of integrations. We use sensor origin, direction and range to describe LiDAR point clouds. With the direction and sensor origin, we sample points and do range integration for the rendering loss. There is no depth maps in NuScenes and Waymo. We utilize the LiDAR point clouds and sensor position to achieve the integration.
>
> **Q3: The rendering loss is mentioned many times in the paper, which should refer to the loss of rendering the point cloud as an image. However, the method introduces the projection of points to pixels through a given projection matrix. The rendering process seems redundant?**
>
> The projection from points to pixels is only used when projecting image features to 3D space for one time, which is similar to [A]. For rendering loss, we render both camera images and LiDAR point clouds, which are two separate and parallel processes. To render camera pixels, we sample rays starting from camera origins and conduct color value integration to predict the pixel values. To render LiDAR point clouds, we sample rays starting from the LiDAR origin and conduct range (depth) integration to predict the LiDAR points.
>
> **Q4: In Formula 1, the author says that $\hat{\mathbf{P}} \in \mathbb{R}^{\hat{D} \times \hat{H} \times \hat{W} \times \hat{d}_{\mathrm{p}}}$ is the generated embedding feature, which is obviously not per-point features. For each superscript, the author should give a clear explanation, and it is recommended to explain it in text and in Figure 2.**
>
> *$f^{enc}_{P}$* first voxelize the raw point cloud and then conduct sparse convolution [B] on the voxels. Then we densify the sparse features to *$\hat{\mathbf{P}}\in\mathbb{R}^{\hat{D}\times \hat{H} \times \hat{W} \times \hat{d}_{\text{P}}}$*, where part of the voxels are empty (zero values). The 3D image feature *$\hat{\mathbf{I}}\in\mathbb{R}^{\hat{D}\times \hat{H} \times \hat{W} \times \hat{d}_{\text{I}}}$* is also sparse. Then we concatenate  *$\hat{\mathbf{P}}$* and *$\hat{\mathbf{I}}$* and use dense 3D convolution *$f^{\text{enc}}_{\text{fusion}}$* to further embed the fused features. We will update the text and figure accordingly.
>
> **Q5: Formula 4 needs more introduction. How are $L_{rend}$ and $L_{proto}$ implemented, and what type of loss is used?**
>
> For *$\mathcal{L}_{\text{rend}}$*, the detailed formulation is described in Formula 10 in Line 261, which is L1 loss. For *$\mathcal{L}_{\text{proto}}$*, the detailed formulation is described in Formula 15 to 18 in Line 328-354. We will add reference for them in Section 3.1.
>
> [A] Liu Z, Tang H, Amini A, et al. Bevfusion: Multi-task multi-sensor fusion with unified bird's-eye view representation[J]. arXiv preprint arXiv:2205.13542, 2022.
> [B] Graham B, Van der Maaten L. Submanifold sparse convolutional networks[J]. arXiv preprint arXiv:1706.01307, 2017.

---

> > ### Author Response · Authors · 2025-11-20
> > **Response to Reviewer tkBv. (Part 2)**
> >
> > **Q6: The example of curvature sampling in Figure 3 is unconvincing to me. Because there is no curvature in a two-dimensional image, it is recommended that the author redraw the drawing as described in the text. Operate on LiDAR point cloud, reflect the signed distance function, surface normal, etc.**
> >
> > Figure 3 serves as a conceptual illustration to convey the intuition behind curvature sampling and the actual curvature computation is performed on the 3D LiDAR point cloud using the signed distance function and surface normals as described in Section 3.3, not on the 2D image itself. We think image is more intuitive for reader and we agree that adding LiDAR point clouds will make it more expressive. We will update the figure and add projected LiDAR points onto the image with SDF/surface normals.
> >
> > **Q7: The experiment section of this paper shows that a pre-trained model is used for fine-tuning, but it is not accurate to call this an unsupervised method.**
> >
> > Firstly, the image encoder is pre-trained in unsupervised manner and the pre-training data has no overlap with those in NuScenes or Waymo. Secondly, it is a common practice in previous literatures including UniPAD, SLidR and PPKT. We think we could call it an unsupervised method.
> >
> > **Q8: Figure 4 obviously needs to be improved. It is recommended that the author add sample sub-images in a row to fill the blanks. In addition, why are there fewer points with high sampling weights and concentrated at the bottom of the car object? According to theoretical analysis, shouldn't the area with large curvature transformation be concentrated above or around the car? The author must conduct sufficient experimental analysis, as I guess this may be a defect caused by the implicit signal function used.**
> >
> > The visualization in Figure 4 is rendered from a Bird's Eye View (BEV) perspective, which may create a visual impression that high-weight points concentrate at the "bottom" of vehicles. However, what appears as "bottom" in BEV actually corresponds to the sides and boundaries around the vehicle in 3D space. We will update Figure 4 using a 3D zoom-in to make the visualization clearer.
> >
> > **Q9: The method of this paper is abbreviated as CLAP (Curvature sampLing and leArnable Prototype). Is it to be as similar as possible to the classic work CLIP? I think it is easier to understand if CSLP appears in the title. Of course, it depends entirely on the author's ideas.**
> >
> > CLIP is used for object-centered image and text contrastive pre-training. CLAP is used for scene point clouds and camera image joint pre-training. The basic ideas are different.
> >
> > **Q10: About symbols, fonts and references.**
> >
> > Thanks for point out and we will add reference discussion and update accordingly.
> >
> >
> > We hope that our replies and experiments address your concern and we are happy for any further discussion.
> >
> > Best,
> >
> > Authors of Submission 1808

---

> > > ### Comment · Reviewer_tkBv · 2025-11-23
> > >
> > > Thanks for your thoughtful response. My concerns have largely been addressed, and I will maintain my positive rating.
> > >
> > > Additionally, I note that the other three reviewers have provided somewhat negative feedback. I hope your response will also gain their approval.
> > >
> > > Best of luck!

---

> > > > ### Author Response · Authors · 2025-11-26
> > > >
> > > > Thank you for your response! We are glad that we resolve your concerns.

---

### Official Review · Reviewer_brLm · 2025-10-18

**Soundness:** 2
**Presentation:** 3
**Contribution:** 2
**Rating:** 4
**Confidence:** 3

**Summary:**

This paper proposes CLAP, an unsupervised 3D representation learning framework that enables joint pre-training of LiDAR and camera encoders via differentiable rendering. It builds upon the UniPAD framework but introduces two key innovations: Curvature Sampling – a memory-efficient sampling strategy that prioritizes points with higher geometric curvature to retain informative structures during rendering. Prototype Learning – a cross-modal feature alignment mechanism using learnable prototypes optimized via Expectation-Maximization (EM), swapping prediction loss, and Gram Matrix Regularization to bridge image and point cloud modalities. Experiments on NuScenes and Waymo demonstrate good improvements over previous SOTA methods, particularly under few-shot fine-tuning settings. Ablation studies further show the effectiveness of each proposed component.

**Strengths:**

- Clear motivation

- Curvature Sampling is intuitive; it prioritizes complex regions (e.g., vehicles, edges) while maintaining low computational overhead in theory.

- Prototype Learning sounds crucial to align and interact between LiDAR and image modalities, improving cross-modal understanding.

- Experiments include both NuScenes (5%) and Waymo (1%), with detailed scaling analyses (0.5–5%) showing gains.

- Ablation results show contributions of each component.

**Weaknesses:**

- The authors still focus mainly on few-shot downstream training (NuScenes 5%, Waymo 1%), which, while useful for showing sample efficiency, is non-standard in representation learning. Full-data fine-tuning results would better demonstrate scalability and practical performance benefits.

- Although Curvature Sampling is motivated as memory-efficient, the paper does not explicitly quantify memory or runtime savings compared to UniPAD’s “Memory-friendly Ray Sampling.”

- The prototype mechanism is interesting, but it’s unclear how well these prototypes correspond to meaningful physical parts or semantic regions. From figure 5, only the prototype assigned to the ground and the car can be figured out correctly.

- The ablation table confirms effectiveness but doesn’t directly compare against UniPAD’s sampling strategies

**Questions:**

Please check the weakness.

---

> ### Author Response · Authors · 2025-11-20
> **Response to Reviewer brLm. (Part 1)**
>
> Dear Reviewer brLm,
>
> Thank you for your precious time on the review and your constructive suggestions to improve our manuscript! We appreciate the acknowledgment that CLAP has clear motivation, Curvature Sampling prioritizes complex regions while maintaining low computational overhead in theory, prototype learning improves cross-modality understanding and experiments show the effectiveness of CLAP.
>
> We provide further discussions on your questions as belows:
>
> **Q1: The authors still focus mainly on few-shot downstream training (NuScenes 5\%, Waymo 1\%), which, while useful for showing sample efficiency, is non-standard in representation learning. Full-data fine-tuning results would better demonstrate scalability and practical performance benefits.**
>
> We use uniform sampling over frame-level to sample the labeled data for fine-tuning. Taking NuScenes as an example, there are 850 scenes and each scene (a sequence of data) has approximately 40 labeled frames. In the 5\% downstream experiments, we uniformly sample 2 frames in each scene and have a total of around 1700 frames for fine-tuning. For 0.5\% experiments, we uniformly sample 1 frame every 5 scenes. And the sampling is done once and fixed for all compared methods to ensure fair comparison. In this way, we preserves the natural distribution of objects and scenes in the dataset as much as possible with much fewer labeled data. We believe this few-shot setting is valuable for several important reasons. Firstly, in practical autonomous driving applications, unlabeled data is significantly more abundant than labeled data. Our setting simulates this realistic scenario where only a small portion of collected data can be affordably annotated. Secondly, this experimental setup is a common practice in previous unsupervised 3D representation learning works like \cite{pointcontrast,mvjar}. Last but not least, we first train the randomly initialized model with increasing iterations until performance plateaus (convergence). Only then do we fine-tune the pre-trained models with this fixed number of iterations. This ensures that our reported improvements reflect genuine performance gains rather than simply faster convergence.
>
> And we also provide experiments on fine-tuning on whole datasets in Appendix B.4. The results are as belows,
>
> |  Init. |  mAP  |  NDS  |  mAP  |  mAPH |
> |:------:|:-----:|:-----:|:-----:|:-----:|
> |  Rand. | 64.57 | 67.61 | 69.00 | 67.57 |
> | UniPAD | 64.69 | 68.06 | 69.03 | 67.56 |
> |  CLAP  | 65.11 | 68.38 | 69.49 | 68.04 |
>
> CLAP improves random initialization by +0.55 whereas UniPAD by +0.15 on average, showing CLAP’s effectiveness. With Table 3 in the paper, more gain can be expected with more unlabeled data in real scenario.
>
> **2: Although Curvature Sampling is motivated as memory-efficient, the paper does not explicitly quantify memory or runtime savings compared to UniPAD’s “Memory-friendly Ray Sampling.”**
>
> Excellent point. We provide rigorous quantitative evidence below:
>
> |       Init.       | GPU Memory | Time/epoch |
> |:-----------------:|:----------:|:----------:|
> | UniPAD (separate) |    ~60GB   |   40 mins  |
> |   UniPAD (joint)  |    ~80GB   |   70 mins  |
> |        CLAP       |    ~40GB   |   50 mins  |
>
> Also in Table 4, we provide comparison between Curvature Sampling and the sampling method applied in UniPAD. The results in Table 4 together with the statistic here show both the effectiveness and efficiency of Curvature Sampling and CLAP.
>
> **Q3: The prototype mechanism is interesting, but it’s unclear how well these prototypes correspond to meaningful physical parts or semantic regions. From figure 5, only the prototype assigned to the ground and the car can be figured out correctly.**
>
> In figure 5, we use the model pre-trained by CLAP to infer the 3D features and assign prototypes to different LiDAR points in the 3D space. Then we use different random colors to indicate different prototypes and visualize them.
>
> It can be found that the background road plane inside the same frame is generally assigned to the same prototype. Meanwhile, walls are assigned to other classes. And different foreground vehicles are assigned to different prototypes. This demonstrates that the proposed prototype learning scheme to some extend learns to represent parts of the scenes with prototypes in an unsupervised manner.
>
> However, as no labels are provided during pre-training, the prototype assignment has some noise, for example some of the road plane points are assigned to other prototypes and some pedestrian points are not that distinguishable from the background.

---

> > ### Author Response · Authors · 2025-11-20
> > **Response to Reviewer brLm. (Part 2)**
> >
> > **Q4: The ablation table confirms effectiveness but doesn’t directly compare against UniPAD’s sampling strategies**
> >
> > As discussed in Line 467-470, We conduct ablation study to evaluate the effectiveness of different components. Results are in Table 4. The first line is separate pre-training with UniPAD (Yang et al., 2024). The second one is jointly pre-training using UniPAD[A] and uniform sampling guided by range (“Memory-friendly Ray Sampling” in [A]) to address the GPU memory limitation.
> >
> > It is a direct comparison against UniPAD’s sampling strategies.
> >
> > We hope that our replies and experiments address your concern and we are happy for any further discussion.
> >
> > Best,
> >
> > Authors of Submission 1808
> >
> > [A] Yang H, Zhang S, Huang D, et al. Unipad: A universal pre-training paradigm for autonomous driving[C]//Proceedings of the IEEE/CVF conference on computer vision and pattern recognition. 2024: 15238-15250.

---

> ### Author Response · Authors · 2025-11-26
> **Look forward to further discussion. Thank you!**
>
> Dear Reviewer brLm,
>
> We hope that our replies and experiments address your concern and we are happy for any further discussion. Look forward to your reply.

---

> > ### Comment · Reviewer_brLm · 2025-11-26
> > **Reply to Authors**
> >
> > Thanks the authors for providing these detailed replies. Most of my concerns are addressed and I would like to upgrade to 6.

---

### Official Review · Reviewer_4dye · 2025-10-25

**Soundness:** 3
**Presentation:** 2
**Contribution:** 3
**Rating:** 4
**Confidence:** 3

**Summary:**

CLAP is an unsupervised pre-training method for fusion 3D perception in autonomous driving, with the core idea of achieving joint unsupervised pre-training of images and LiDAR point clouds through **curvature sampling** and **learnable prototype learning**.

Firstly, CLAP uses a curvature sampling strategy to select informative points and pixels, solving the computational cost problem of multi-modal data processing. Secondly, it randomly initializes learnable prototypes and combines the Expectation-Maximization (EM) algorithm to construct a cross-modal common feature space, associating image semantics with point cloud geometric information. Meanwhile, it designs a swapping prediction loss to explore inter-modal interaction and introduces Gram matrix regularization to avoid prototype collapse.

Experiments on NuScenes and Waymo datasets show that CLAP performs excellently in downstream tasks such as 3D object detection (e.g., BEVFusion, CenterPoint) and semantic segmentation (e.g., Cylinder3D), with significant performance improvements compared to random initialization and existing SOTA methods (e.g., UniPAD). It also has outstanding scalability in few-shot fine-tuning scenarios and can effectively improve model performance during cross-dataset transfer.

**Strengths:**

1.Strong scalability of the method: In few-shot fine-tuning scenarios, CLAP’s performance improvement increases as the amount of training data decreases, demonstrating strong scalability potential.

2.Practical engineering significance: CLAP reduces the computational cost of joint unsupervised pre-training for image and point cloud modalities, facilitating information interaction between different modalities.

3.Experimental validation of effectiveness: On benchmarks (e.g., NuScenes, Waymo), CLAP outperforms existing baseline methods. Cross-task (3D object detection, semantic segmentation) transfer experiments further validate its effectiveness.

**Weaknesses:**

1.Insufficient analysis of results:Further analysis is needed to determine whether CLAP’s advantages lie in handling complex scenarios or achieving broad accuracy gains. Notably, its inferior performance on specific classes (vs. UniPAD/PPKT in Table 1) investigating the underlying causes.

2.Memory efficiency requires further proof:UniPAD separately pre-train the image and point cloud encoders due to GPU memory constraints. Although curvature sampling is introduced to mitigate this issue, the absence of rigorous empirical validation (e.g., memory reduction metrics or training efficiency analysis) limits its persuasiveness. Future work should include quantitative evaluations to demonstrate its benefits.

**Questions:**

Given that learnable prototypes are employed to represent 3D scene segments, with similarity estimated via the Gram matrix $G$ to prevent collapse, have you conducted ablation studies on $N_{K}$? Does there exist an optimal $N_{K}$? beyond which the additional prototypes become noise sources rather than meaningful representations?

---

> ### Author Response · Authors · 2025-11-20
> **Response to Reviewer 4dye.**
>
> Dear Reviewer 4dye,
>
> Thank you for your precious time on the review and your constructive suggestions to improve our manuscript! We appreciate the acknowledgment that CLAP has strong scalability, practical significance and experiments validate the effectiveness of CLAP.
>
> We provide further discussions on your questions as belows:
>
> **Q1: Insufficient analysis of results:Further analysis is needed to determine whether CLAP’s advantages lie in handling complex scenarios or achieving broad accuracy gains. Notably, its inferior performance on specific classes (vs. UniPAD/PPKT in Table 1) investigating the underlying causes.**
>
> Thank you for the insightful suggestion! CLAP shows broad accuracy gains across most categories with particularly strong improvements in geometrically complex objects, including Construction Vehicle (+1.67\%), Bus (+7.36\%), Bicycle (+6.58\%) and Truck (+5.62\%). Meanwhile, CLAP performs worse on traffic cones, trailer and motorcycle. The reason are two parts.
>
> First of all, for traffic cones, the geometry is too simple and one of the pre-training goal of CLAP is to reconstruct the scene geometry. Thus for category with simple geometry, pre-training benefit more from semantic distillation method like PPKT. It can be found that PPKT excels both UniPAD and CLAP on traffic cone.
>
> Secondly, trailer is less distinguishable in geometry and motorcycle is similar to bicycle. This makes CLAP performs a bit worse than UniPAD and PPKT.
>
> Although CLAP performs inferior to UniPAD/PPKT on specific classes, CLAP achieves consistent improvement over random initialization over all categories which UniPAD/PPKT cannot achieve. And for overall performance on mAP and NDS, CLAP achieves +2.48\% mAP overall, indicating broad accuracy gains.
>
> **Q2: Memory efficiency requires further proof:UniPAD separately pre-train the image and point cloud encoders due to GPU memory constraints. Although curvature sampling is introduced to mitigate this issue, the absence of rigorous empirical validation (e.g., memory reduction metrics or training efficiency analysis) limits its persuasiveness. Future work should include quantitative evaluations to demonstrate its benefits.**
>
> Excellent point. We provide rigorous quantitative evidence below:
>
> |       Method       | GPU Memory | Time/epoch |
> |:-----------------:|:----------:|:----------:|
> | UniPAD (separate) |    ~60GB   |   40 mins  |
> |   UniPAD (joint)  |    ~80GB   |   70 mins  |
> |        CLAP       |    ~40GB   |   50 mins  |
>
> Also in Table 4, we provide comparison between Curvature Sampling and the sampling method applied in UniPAD. The results in Table 4 together with the statistic here show both the effectiveness and efficiency of Curvature Sampling and CLAP.
>
> **Q3: Given that learnable prototypes are employed to represent 3D scene segments, with similarity estimated via the Gram matrix
>  to prevent collapse, have you conducted ablation studies on NK Does there exist an optimal NK beyond which the additional prototypes become noise sources rather than meaningful representations?**
>
> This is an insightful question about the optimal prototype configuration. A larger $N_K$ will make training harder and might bring too much noise while a smaller $N_K$ might not be enough to cover the scene semantics during pre-training. We further conduct parameter sensitivity experiments on $N_K$. The results are as belows:
>
> | $N_K$ |  mAP  |  NDS  |
> |:----:|:-----:|:-----:|
> |  256 | 50.89 | 56.78 |
> |  512 | 51.17 | 57.04 |
> | 1024 | 50.93 | 56.85 |
>
> We hope that our replies and experiments address your concern and we are happy for any further discussion.
>
> Best,
> Authors of Submission 1808

---

> ### Author Response · Authors · 2025-11-26
> **Look forward to further discussion. Thank you!**
>
> Dear Reviewer 4dye,
>
> We hope that our replies and experiments address your concern and we are happy for any further discussion. Look forward to your reply.

---

> > ### Comment · Reviewer_4dye · 2025-11-26
> >
> > I appreciate the authors' detailed responses to each question, which effectively addressed my concerns. Therefore, I am willing to raise my score.

---

### Official Review · Reviewer_JtoM · 2025-10-31

**Soundness:** 3
**Presentation:** 3
**Contribution:** 2
**Rating:** 4
**Confidence:** 5

**Summary:**

This paper proposes CLAP, a framework for unsupervised fused joint pre-training of LiDAR and camera data through differentiable rendering. CLAP incorporates two key components: a Curvature Sampling strategy to select geometrically informative regions and a set of learnable prototypes to represent distinct parts of a 3D scene, which are optimized using an Expectation-Maximization (EM) procedure.

**Strengths:**

The manuscript is well-written, and the motivation for the work is clearly established.

The authors design a curvature sampling strategy to identify informative points and pixels for sampling.

Learnable prototypes are utilized to establish a common feature space, and an Expectation-Maximization approach is employed to optimize these prototypes, enabling them to represent distinct parts of the 3D scene.

**Weaknesses:**

The coverage of related work and compared methods lacks comprehensiveness. Notably, several recent methods from the "3DTrans" GitHub repository, which reportedly achieve strong performance, are neither discussed nor included in the experimental comparisons.

When compared to other outdoor self-supervised learning (SSL) methods that also employ differentiable rendering, the primary novelty of the proposed CLAP framework appears to be the Curvature Sampling strategy.

The use of prototype learning is a well-established technique in the SSL community. Consequently, positioning this as a major contribution is not fully persuasive, as its incremental advancement over existing approaches is not sufficiently demonstrated.

**Questions:**

I recommend expanding the comparative analysis to include several recent methods from the '3DTrans' benchmark. A fair and thorough comparison is essential to validate the claim that CLAP represents a meaningful advancement over existing alternatives. The current evaluation feels selective and does not fully establish the method's state-of-the-art status.

I would like the authors to report the mean and variance of five replicates in all experiments, as reproducibility has always been a problem in this field.

I'd like to know how the prototype learning proposed in this paper differs from other existing prototype learning methods. If it's merely being borrowed from this field, then its value wouldn't be that great.

How much time does online differentiable rendering consume? As far as I know, it slows down the pre-training process. Please explain this in detail.

---

> ### Author Response · Authors · 2025-11-20
> **Response to Reviewer JtoM. (Part1)**
>
> Dear Reviewer JtoM,
>
> Thank you for your precious time on the review and your constructive suggestions to improve our manuscript! We appreciate the acknowledgment that the paper is well-written, motivation is clear and the curvature sampling strategy helps identify informative points and pixels for sampling.
>
> We provide further discussions on your questions as belows:
>
> **Q1: The coverage of related work and compared methods lacks comprehensiveness. Notably, several recent methods from the "3DTrans" GitHub repository, which reportedly achieve strong performance, are neither discussed nor included in the experimental comparisons.**
>
> We appreciate the reviewer pointing out the 3DTrans repository. However, we respectfully clarify that methods in 3DTrans focus on supervised/semi-supervised pre-training, which requires labels during pre-training, and domain adaptation, which conducts supervised training on one dataset and transfers to other datasets. The focus of 3DTrans is fundamentally different problem settings from our work. The baseline methods in our experiments (ALSO, OCC-MAE, SLidR, PPKT, UniPAD) represent the current SOTA in unsupervised 3D pre-training with LiDAR and camera modalities, which is the specific problem we address.
>
> Meanwhile, we agree that including supervised/semi-supervised pre-training methods would enrich our analysis. We provide additional experiments using supervised/semi-supervised methods from 3DTrans (ADPT and SPOT).
>
> |     Init.    |  mAP  |  NDS  |
> |:------------:|:-----:|:-----:|
> | From-scratch | 48.69 | 55.28 |
> |     SPOT     | 50.98 | 57.20 |
> |     ADPT     | 51.05 | 56.25 |
> |     CLAP     | 51.17 | 57.04 |
>
> These results show that CLAP remains competitive even when compared with methods that use labels during pre-training, demonstrating the effectiveness of our unsupervised approach.
>
> **Q2: When compared to other outdoor self-supervised learning (SSL) methods that also employ differentiable rendering, the primary novelty of the proposed CLAP framework appears to be the Curvature Sampling strategy.**
>
> We would like to clarify that CLAP represents a substantial departure from UniPAD, which employs differentiable rendering for pre-training. While both methods use neural-rendering decoders (a common approach in the field), our decoder differs by incorporating an additional surface signed distance field loss for better scene geometry optimization. The fundamental distinction is that UniPAD performs separate pre-training of each modality, while CLAP enables joint pre-training with Curvature Sampling and introduces cross-modal interactions via Prototype Learning, which are paradigm shift. CLAP's novel contributions (Curvature Sampling, Prototype Learning with EM training, Swapping prediction loss, and Gram Matrix Regularization) have no counterparts in UniPAD and enable capabilities that UniPAD cannot achieve, such as knowledge transfer between modalities during pre-training. Our ablation study in Table 4 quantifies this difference.
>
> **Q3: The use of prototype learning is a well-established technique in the SSL community. Consequently, positioning this as a major contribution is not fully persuasive, as its incremental advancement over existing approaches is not sufficiently demonstrated.**
>
> We acknowledge that prototype learning is established in the SSL community, but we emphasize that our contribution is not simply applying existing prototype methods, but rather novel adaptations for 3D scene understanding:
>
> (1) Different purpose: Unlike SwAV [1], where prototypes represent different categories within the same modality (images), our prototypes represent parts of 3D scenes and bridge two different modalities (LiDAR + camera).
>
> (2) Expectation-Maximization training scheme (Section 3.4, Eq. 15): We propose maximizing similarity between 3D embeddings and prototypes by minimizing entropy of assignment probabilities - designed specifically for 3D scene part representation, not category clustering.
>
> (3) Gram Matrix Regularization (Eq. 17): A novel regularization term to prevent prototype collapse in the 3D multimodal setting, which differs from the Sinkhorn-Knopp constraints used in SwAV.
>
> (4) Cross-modal swapping prediction (Eq. 16): While inspired by SwAV, our swapping operates between different modalities (predicting camera prototypes from LiDAR embeddings and vice versa) rather than different views of the same modality.
>
> Empirical validation: Table 4 shows that prototype learning provides improvement beyond curvature sampling alone, and the combination achieves 100\% more improvement than SOTA (2.48\% vs. 1.12\% from UniPAD).
>
> [A] Caron M, Misra I, Mairal J, et al. Unsupervised learning of visual features by contrasting cluster assignments[J]. Advances in neural information processing systems, 2020, 33: 9912-9924.

---

> ### Author Response · Authors · 2025-11-20
> **Response to Reviewer JtoM. (Part2)**
>
> **Q4: I would like the authors to report the mean and variance of five replicates in all experiments, as reproducibility has always been a problem in this field.**
>
> We appreciate the insightful suggestions from reviewer. First of all, we use the same fixed random seed for all experiments in the main paper for reproducibility. Secondly, we indeed provided experiment results for repeated evaluation in appendix B.2 to further reveal the training robustness. We repeat fine-tunings on NuScenes for 5 times with random initialization, UniPAD and CLAP. Mean and standard deviation of mAP are 48.55 ± 0.18\% (Rand.), 49.66 ± 0.29\% (UniPAD) and 51.16 ± 0.10\% (CLAP). CLAP achieves the best average performance and robustness against random seeds.
>
> **Q5: How much time does online differentiable rendering consume? As far as I know, it slows down the pre-training process. Please explain this in detail.**
>
> Thanks for the insightful question! As CLAP needs to render both RGB pixels and LiDAR points during pre-training, CLAP requires approximately 25\% more than previous methods per epoch (50 mins v.s. 40 mins on 8-H100) for pre-training. If comparing to UniPAD, which requires separate pre-training for image and LiDAR encoders, CLAP even saves 37\% during pre-training.
>
> We hope that our replies and experiments address your concern and we are happy for any further discussion.
>
> Best,
> Authors of Submission 1808

---

> ### Author Response · Authors · 2025-11-26
> **Look forward to further discussion. Thank you!**
>
> Dear Reviewer JtoM,
>
> We hope that our replies and experiments address your concern and we are happy for any further discussion. Look forward to your reply.

---

> > ### Comment · Reviewer_JtoM · 2025-11-27
> > **Official Comment**
> >
> > Thanks for the detailed response. My questions have been addressed and have thus raised the rating to 6.

---

### Author Response · Authors · 2025-12-03
**Summary of rebuttal. Thank you!**

## Rebuttal Summary
We sincerely thank all reviewers and the Area Chair for the thorough reviews and constructive feedback. We are grateful for the time and effort invested in evaluating and discussing our work. We thanks the reviewers' acknowlegdement that (1) the paper is well-written (Reviewer JtoM and tkBv), (2) the motivation is clear (Reviewer JtoM, brLm and tkBv), (3) CLAP has obvious practical significance (Reviewer 4dye and tkBv), (4) Curvature Sampling helps identify informative region for sampling (Reviewer JtoM and brLm), (5) the paper is highly complete and the methods are sufficiently original (Reviewer tkBv) and (6) experiment results demonstrates the effectiveness of CLAP (Reviewer 4dye, brLm and tkBv).

During the rebuttal, we addressed reviewers' concerns by additional experiments and enhanced analysis, which we discuss as follows.

### Additional Experiments

(1) Memory and time efficiency analysis (Reviewer JtoM, 4dye and brLm): We provide quantitative metrics showing memory reduction and training speedup.

(2) Prototype number ($N_K$) ablation (Reviewer 4dye): We conduct parameter sensitivity experiments on the prototype number selection, which demonstrates the optimal $N_K$ = 512.

(3) Comparison to Semi-supervised Pre-training methods from 3DTrans (Reviewer JtoM). We conduct experiments comparing to supervised and semi-supervised pre-training methods AD-PT and SPOT in the 3DTrans repo. Results show that our unsupervised pre-training method CLAP remains competitive even when compared with methods that use labels during pre-training.

(4) Repeated Experiments (Reviewer JtoM). Actually we provide it in appendix B.2 and CLAP achieves the best average performance and robustness against random seeds.

(5) Fine-tuning on more data (Reviewer brLm). We provide the results in appendix B.4 and CLAP improves random initialization by +0.55 whereas UniPAD by +0.15 on average, showing CLAP’s effectiveness. With Table 3 in the paper, more gain can be expected with more unlabeled data in real scenario.

### Enhanced Analysis Provided:

(1) Difference between CLAP and UniPAD (Reviewer JtoM): We provide further analysis and explanation on the difference between CLAP and UniPAD, highlighting CLAP's novel contributions (Curvature Sampling, Prototype Learning with EM training, Swapping prediction loss, and Gram Matrix Regularization).

(2) Discussion on Prototype Learning scheme (Reviewer JtoM): We discuss the different scenarios between previous prototype learning works and CLAP, making it clearer that CLAP provides a novel adaptation of prototype learning for 3D scene understanding.

(3) Detailed results analysis (Reviewer 4dye): First of all, CLAP shows broad accuracy gains across most categories with particularly strong improvements in geometrically complex objects. Although CLAP achieves slightly worse performance for geometrically simple and less distinguishable objects, CLAP achieves consistent improvement over random initialization over all categories which UniPAD/PPKT cannot achieve and the best overall performance.

(4) Discussion on the visualization results on prototype learning (Reviewer brLm): Although the prototype assignment has some noise due to that no label is available during pre-training, CLAP successfully assign background and different foreground vehicle into different prototypes, demonstrating the effectiveness of the prototype learning scheme.

(5) About the ablation study (Reviewer brLm): The results in Table 4 are direct comparison against UniPAD’s sampling strategies, showing the effectiveness of Curvature Sampling strategy.

(6) Clarification on Figure 1, 2, 3 and 4 (Reviewer tkBv): we provide further explanation on each figure to resolve the confusion.

(7) Clarification on network architecture (Reviewer tkBv): we provide further explanation on the 3D voxelization, sparse convolution and projection from 2D features to 3D space to deal with the concern.

### Summary of the discussion period.

In the early discussion phase (prior to the data exposure), all the reviewers participated in the discussion. Reviewer JtoM and 4dye replied that all the concerns are addressed and they **updated** the scores from 4 to 6. Reviewer brLm stated that most of the concerns are resolved and **updated** the score from 4 to 6. For Reviewer tkBv, most of the concerns are resolved and score is maintained as 8. In summary, all reviewers show a consistent inclination toward acceptance.

Thank you again for your precious time and efforts during review and rebuttal.

---

### Meta-Review · Area_Chair_FYNR · 2025-12-28

**Summary:**

This paper presents a joint unsupervised differentiable-rendering-based pre-training method for images and point clouds. All reviewers have engaged in the discussion and acknolwdged the increase of the ratings to 6 or above. Authors should ensure the revisions are properly included into the revised version.

Some specific concerns are: comparative analysis to include several recent methods from the '3DTrans' benchmark. How the prototype learning proposed in this paper differs from other existing prototype learning methods. More ablation studies should be conducted. The paper does not explicitly quantify memory or runtime savings compared to existing methods.

**Reviewer Scores:**

it seems final ratings cannot be viewed by area chairs. Reviewers have confirmed their increase of ratings thus the paper has been recommended for acceptance.

---

### Decision · Program_Chairs · 2026-01-26

Accept (Poster)